# The p53 and Calcium Regulated Actin Rearrangement in Model Cells

**DOI:** 10.3390/ijms23169078

**Published:** 2022-08-13

**Authors:** Alexandra Hencz, Edina Szabó-Meleg, Muhammad Yaqoob Dayo, Ardora Bilibani, Szilvia Barkó, Miklós Nyitrai, Dávid Szatmári

**Affiliations:** 1Medical School, Department of Biophysics, University of Pécs, H-7624 Pécs, Hungary; 2Medical School, Institute of Physiology, University of Pécs, H-7624 Pécs, Hungary

**Keywords:** Ca^2+^, actin, p53, GSN, JMY

## Abstract

Long-term cellular stress maintains high intracellular Ca^2+^ concentrations which ultimately initiates apoptosis. Our interest is focused on how the gelsolin (GSN) and junctional mediating and regulating Y protein (JMY) play important roles in stress response. Both of these proteins can bind p53 and actin. We investigated using in vitro fluorescence spectroscopy and found that the p53 competes with actin in GSN to inhibit p53–JMY complex formation. A high Ca^2+^ level initializes p53 dimerization; the dimer competes with actin on JMY, which can lead to p53–JMY cotransport into the nucleus. Here we investigated how the motility and division rate of HeLa cells changes due to low-voltage electroporation of GSN or JMY in scratching assays. We revealed that JMY inhibits their motion, but that it can accelerate the cell division. GSN treatment slows down cell division but does not affect cell motility. HeLa cells fully recovered the gap 20 h after the electroporation with JMY and then started to release from the glass slides. Taken together, our in vitro results indicate that GSN and JMY may play an important role in the cellular stress response.

## 1. Introduction

The cytoskeletal system determines the shape of the cell, but it is also involved in intercellular connections, cell motility, division, and contraction, and in addition, it plays an important role in intracellular transport and signalization [1]. The cellular stress response is a wide range of molecular changes that cells undergo in response to environmental stressors, including extremes of temperature, exposure to toxins, DNA damage, or mechanical effects. The various processes involved in cellular stress responses serve the adaptive purpose of protecting a cell against unfavorable environmental conditions, both through short-term mechanisms that minimize acute damage to the cell’s overall integrity and through longer-term mechanisms that provide the cell a measure of resiliency against similar adverse conditions [2]. A primary stress response is Ca^2+^ influx caused by protein kinase activity [3], then among other things, the microfilaments’ quick remodeling can change the cell motility, division, and intracellular transport. Long-term cellular stress maintains high cytoplasmic Ca^2+^ concentrations, leading to modified physiology of the cell and finally apoptosis [4]. A high number of studies are related to the direct effect of tumor suppressor and apoptotic factors on the cytoskeletal system, dynamics of microfilaments and actin-binding proteins [5,6,7,8,9]. However, it is not clear yet how the Ca^2+^ signal leads to the signaling of the apoptotic factors [10] or how they affect the rearrangement of the cytoskeletal elements during apoptosis. In our study, we focused on two important actin-binding proteins: gelsolin (GSN) [11] and junctional mediating and regulating Y protein (JMY) [12]. Both can bind tumor suppressor p53, and it was also observed that GSN–p53 was localized near the nuclear envelop; the JMY–p53 showed a distribution between cytoplasm and nucleoplasm [11,13]. The complexes of GSN, JMY and p53 are possibly regulating elements of cytoskeletal remodeling and motility linked to the transformation of dividing cells. GSN and JMY proteins play a relevant role in tumorigenesis and invasion. The overexpression of GSN depends on the type of the cells and can induce [14,15,16] or reduce [17,18,19] their motility and division; the absence of protein can initialize the cancer formation [20,21]. However, with gene silencing, it enhances the epithelial-mesenchymal transformation of cells [22], but in some cases, it shows multiple expression patterns during tumorigenesis [23,24]. All are based on the calcium-activated actin severing, nucleating, and capping functions of GSN, and can also be related to the inactivation of GSN by PIP2 binding with some sensitivity to the cytoplasmic nucleotide level [25,26,27,28,29,30,31]. The JMY is a proapoptosis factor; it can be transported into the nucleus where it enhances the transcription of the p53 gene [25]. However, in the cytoplasm, JMY arrests the E-cadherin exhibition, leading to a high number of extracellular interactions and thus quick migration of metastatic cells based on JMY binding to the pointed end of actin filaments and helping the network-building machinery of Arp2/3 complexes [12,13,25,26,27]. Lymph node, primary colorectal, and head–neck carcinoma cells show high, but breast carcinoma cells show low expression of JMY [28].

Interestingly, p53 can have a direct effect on cytoskeletal remodeling. It was previously published that p53 can bind to actin monomers [29] and, in the presence of calcium, can bind to filamentous actin [30]. High-affinity DNA binding of p53 requires p53–tetramers, which can be produced by the dimerization of p53–dimers most typically in the nucleoplasm [31]. The steps of p53 tetramerization are not understood yet, but as structural studies predicted [32,33], the dimer formation is refining the most stable structure of p53. Several studies simply explain the cytoplasmic function of p53, as monomers [34] and lack of function type of the mutations helps the cytoplasmic depletion of dimers [35,36]. It seems that dimers are more frequent in cytoplasm and the nucleoplasmic function of p53 requires the formation of tetramers. As a previous study explained [37], the reason is that p53 phosphorylation is required for the stability of tetramers.

The binding of JMY by three WH2 (Wiskott–Aldrich syndrome protein homology 2 domains) domains to actin depends on the bound nucleotide type, therefore it shows strong interaction with ATP–actin and lower affinity to ADP–actin [38]. The actin nucleator and sequestering functions of JMY are based on the ATP and Ca^2+^ concentration-dependent binding to actin of the three WH2 domains [13,39,40,41,42]. JMY transport into the nucleus is related to p53 binding [43,44] and is lacked by actin [40].

It is very important to reveal how p53 and GSN/JMY are implicated in the stress response and cytoskeletal rearrangement of dividing cells. An interesting question is what kind of mechanism transports GSN from the plasmamembrane to the nuclear envelop and what the factor is which regulates JMY entry into the nucleus. The most interesting question is whether there is any direct relationship between malignant cytoskeletal transformation and malfunctions of the normal cellular stress response. Also, how can GSN and JMY as cytoplasmic or cytoplasmic–nuclear factors link the cytoskeletal remodeling and motility change to the cytoplasmic level of p53? The present study was designed to investigate the direct effect of p53 on actin and the actin–GSN and actin–JMY complexes at different calcium concentrations in order to get insight into the actin rearrangement of stressed cells. Here we are modeling the role of the high cytoplasmic GSN or JMY level in model cells.

## 2. Results

### 2.1. p53 Forms Dimers and Binds to Actin Monomers and Filaments

The intrinsic tryptophan residues fluorescence emission of p53 (0.5 μM) was decreased due to increased Ca^2+^ concentration (Figure 1A). It was observable along with an increased steady-state anisotropy (from 0.066 ± 0.012 to 0.104 ± 0.008) (Figure 1B). The decreasing emission can be interpreted as a structural change to p53 which results in a buried state of tryptophan residues, thus decreasing their fluorescence emission. The increased anisotropy can be described by slower thermal rotational motion of the protein. If tryptophan residues were in turn to be buried in the structure of p53, which, combined with a slower rotational motion of the whole protein, implies a process of structural stabilization with a size increment, this observation could be interpreted as a calcium-induced dimerization of p53. The functional importance of the p53 dimerization could be the initial step for actin binding; thus, we measured the emission of pyrene-labeled actin (2 μM, 5% labeled by pyrene) under polymerizing conditions (100 mM KCl, 1 mM MgCl_2_) in the presence of p53 (0.05 μM). The fluorescence emission of pyrene shows a 4% and 20% increasein the presence of p53 under micromolar and millimolar calcium (Figure 1C), respectively. It seems that millimolar calcium-induced p53 dimer formation and binding to filamentous actin was more expressed than micromolar calcium.

### 2.2. JMY Reorders Actin Filaments by Millimolar Calcium

The in vitro effect of p53 on actin in the presence of JMY is not clear yet. Pyrene–actin (2 μM, 5% labeled by pyrene) was polymerized (by 100 mM KCl, 1 mM MgCl_2_) in the presence of JMY (0.1 μM) to examine the pyrene emission response to p53 (0.1 μM) in the presence or absence of millimolar Ca^2+^ (Figure 2A). The emission of pyrene was elevated by 10% due to p53 then subsequently decreased by 30% by the addition of 1 mM Ca^2+^. However, in the absence of p53, JMY–actin shows a bit higher sensitivity to millimolar Ca^2+^, since the emission was reduced by 40%, while in the absence of JMY and p53, it was only 7%. The JMY shifts the actin monomer/polymer ratio towards the monomers in the presence of millimolar Ca^2+^. The co-sedimentation of filamentous actin (2 μM) (Figure 2B) with p53 (0.1 μM) and/or JMY (0.1 μM) (by SDS-PAGE Appendix A) shows a remarkable loss of filamentous actin: 30%. It reduced the pellet size in favor of supernatant in the presence of 1 mM Ca^2+^ and JMY, suggesting that JMY was sequestering monomeric actin in the presence of millimolar Ca^2+^. The p53 binding caused by minor release (20%) of actin monomers is not yet understood, but the p53 binds directly to filamentous actin (Kd is 10 μM [30]) in the presence of Ca^2+^. Interestingly, with normalized fluorescence emission change of 0.5 μM, p53Alexa488 shows high affinity to the GSN (Kd= 0.207 ± 0.15 μM) (Appendix A), similarly to p53 dimers in the presence of Ca^2+^ (Kd= 0.257 ± 0.067 μM) (Appendix A). The p53 can bind to JMY in the presence (Kd= 0.362 ± 0.081 μM) but cannot in the absence of Ca^2+^ (Appendix A). The formation of actin–GSN or actin–JMY complexes might be modulated by p53 binding. The effect of p53 binding on the formation of these complexes can be measured by the fluorescence anisotropy change of Alexa488-labeled GSN. The steady-state anisotropy change of GSN–Al488 indicates thermal rotational motion, and reveals Ca^2+^-caused structural changes and p53 binding, or by the severing and capping activity of GSN on short actin filaments (under polymerizing conditions 100 mM KCl, 1 mM MgCl_2_) in the presence of millimolar Ca^2+^ with JMY and/or p53 (Figure 2C). The anisotropy of 0.2 μM Alexa488–GSN in the presence of millimolar Ca^2+^ and 0.1 μM p53 (0.109 ± 0.0014) was at the same level as after the addition of 0.2 μM JMY (0.108 ± 0.0022). However, the subsequent addition of filamentous actin (2 μM) increased it a bit (0.115 ± 0.0011) and shows a relatively high value in the absence of JMY and p53 (0.138 ± 0.006). It seems that the JMY and p53 can reduce the length and number of filaments, and thus, GSN can bind only short oligomers and monomers, which was manifested in a small change of anisotropy in the presence of millimolar Ca^2+^.

### 2.3. Actin Competes with p53 on JMY and p53 Competes with Actin on GSN in a Calcium-Dependent Manner

To understand the complex formation-modulating role of p53 on actin–GSN or actin–JMY complexes, we first measured the emission changes of Alexa488-labeled p53 (Figure 3A). As we observed the emission of 0.5 μM p53Alexa488 in the absence of polymerizing salts but in the presence of 1 mM Ca^2+^ was reduced by 3% with the addition of 1 μM actin then by 3% or 10% more by the subsequent addition of 0.5 μM JMY or 0.5 μM GSN, respectively. The emission remains the same with the addition of JMY in the presence of actin and GSN; however, it was reduced by 7% due to GSN in the presence of actin and JMY. This can be explained as that the 1 mM Ca^2+^-provided p53 dimers were able to bind actin and GSN. The GSN-trapped p53 and p53–actin did not react to the JMY addition. However, JMY was able to bind p53 and p53–actin, but they were collected by the subsequent addition of GSN.

The fluorescence emission of 0.5 μM JMYAlexa488 in the absence of polymerizing salts and in the presence of 1 mM Ca^2+^ was reduced by 11% due to 1 μM actin and then increased by 5% due to a subsequent addition of 0.5 μM p53 (Figure 3B). For comparison, under the same conditions, the fluorescence emission of 0.5 μM GSNAlexa488 was reduced by 13% by 1 μM actin and then decreased 10% more by subsequently adding 0.5 μM p53. The emission of JMYAlexa488 was reduced by 12% by 0.5 μM GSN in the presence of actin and p53; however, the emission of GSNAlexa488 was increased by 8% due to 0.5 μM JMY in the presence of actin and p53. We can assume that actin binding can reduce the emission of JMYAlexa488, and then the p53 dimer can remove the actin from the JMY–actin complexes, and thus actin–p53 can be trapped by GSN. If the p53 dimer binds to GSN–actin, subsequently adding JMY can collect actin monomers.

Pyrene–actin (2 μM, 5% labeled by pyrene) was polymerized (100 mM KCl, 1 mM MgCl_2_) in the presence of 0.5 μM GSN (Figure 3C). The emission of pyrene was a good signal to study the response of the actin–GSN complex to the addition of 0.5 μM JMY in the presence or absence of 0.5 μM p53 under different Ca^2+^ concentrations. It was the most sensitive in the presence of micromolar Ca^2+^, where the emission was increased by 12% and then 4% more due to the addition of p53 followed by JMY. In the presence of nanomolar Ca^2+^, the emission was increased by only 4 % and then 4 % more. In the presence of millimolar Ca^2+^ and p53, only the JMY addition caused a minor 4% increase of it. However, in the absence of p53, the emission of pyrene–actin was increased by 12% due to the JMY addition under micromolar Ca^2+^ and by 3 or 6% under nanomolar or millimolar Ca^2+^, respectively. Although, under nanomolar Ca^2+^, the inactive GSN [45] (in the absence of PIP2) can possibly bind actin monomer [46,47] with a minor response to p53 and then JMY addition. Under micromolar Ca^2+^ concentration, the actin filaments are capped by partially active GSN [48,49,50,51] and then can be uncapped by p53 addition. Interestingly, JMY can possibly bind to both uncapped and capped filaments. Under millimolar Ca^2+^ concentrations, active GSN is continuously severing and capping filaments independently of p53, and then JMY can bind to the pointed ends [41] of short filaments.

If pyrene–actin was polymerized in the presence of 0.5 μM JMY (Figure 3D), the emission of pyrene was increased by p53 in a Ca^2+^ concentration-dependent manner (nanomolar 0%, micromolar 4% and millimolar 10%) then a subsequently added 0.5 μM GSN reduced the pyrene emission with 2% under micromolar and 20% under millimolar Ca^2+^. Of interest, in the absence of p53, the emission of JMY-bound pyrene–actin was increased by 4% by the inactive GSN under nanomolar Ca^2+^ and decreased by 5% and 15% by the active GSN under micromolar and millimolar Ca^2+^, respectively. We can suppose that the JMY–actin complex shows a Ca^2+^-dependent response to p53 which can be modified by the capping and/or severing activity of GSN. The p53 possibly competes with actin monomers on GSN under nanomolar Ca^2+^.

### 2.4. Increased Cytoplasmic GSN Level in HeLa Cells

Previous studies published the benefits of low-voltage electroporation [52,53,54,55], as low voltage causes minor stress on cells with a high efficiency of electroporation. We were using current impulses on HeLa cells to increase the GSN concentration in their cytoplasm in order to model the response of stressed cells, as they can increase the expression of stress proteins. The cell size did not show any difference through electroporation with buffer only (Figure 4A–C), but the cell size was reduced by electroporation with 125 nM GSN (Figure 4D–F). The average cell size was significantly decreased by the applied GSN electroporation (Figure 4G). Figure 4H shows that the fluorescent GSN–Alexa568 was localized in the cells, meaning that we successfully applied our low-voltage electroporation system to HeLa cells. A high level of cytoplasmic GSN can be massed next to the cytoplasmic surface of the nuclear envelop in the same localization as in the case of a previous study [11].

### 2.5. High Cytoplasmic GSN or JMY Affects the Division and Motility of HeLa Cells

The aim of our in vitro cellular assay was to increase the cytoplasmic GSN or JMY level in HeLa cells and to examine how they modify the division and motility of the cells. We carried out scratching assays to examine the rate of gap size recovery (Figure 5A–C) by measuring the change to cell motility in the first 5 h and then the cell division rate 12 and 24 h after the short stress of the electroporation. We measured the gap recovery in four cases, in the first case after the simple addition of buffer only (untreated control), in the second case after the addition of proteins (GSN or JMY) in the final 125 nM concentration, in the third case after the electroporation with buffer only (electroporated control), and in the fourth case after the electroporation with the final 125 nM concentration of proteins GSN or JMY in the medium. The rate of recovery depends on the initial size of the gap and the size of the cells, both of which parameters were used to normalize the change of the gap size (Figure 5D) for the analysis of the cellular motility and division rates in cells/hour. The rate differences show that the electroporation by itself can stimulate the cells leading to faster movement in the first 5 h. After the electroporation with JMY, the speed of the motility and division of cells shows interesting kinetics; in the first 5 h they were slowed down, then in the next 19 h they were divided faster than electroporated control cells, ultimately showing the same rate. After electroporation with GSN, the cells were moving identically to untreated control cells, but they ultimately showed a markedly slow rate in their division (Figure 5F). HeLa cells fully recovered the gap 20 h after the electroporation with JMY, then cells started to release from the glass slides (Figure 5G,H).

## 3. Discussion

Here we have demonstrated that the actin rearrangement depends on the complexes of p53, GSN, and JMY. Interestingly, in the presence of millimolar Ca^2+^, the p53 is able to form dimers (Figure 1A,B) and thus can possibly bind to JMY or actin (Figure 1C; Figure 2A,B; Figure 3). GSN can bind p53 independently of the Ca^2+^ level (Figure 3; Appendix A). The free p53 and p53 in a complex with an actin monomer can also be trapped by GSN independently of JMY (Figure 3). We suggest that the JMY was able to bind p53 dimers and p53–actin complexes, but the subsequently added GSN collected the main part of p53 dimers (Figure 2; Figure 3A,B). In our interpretation, the p53 dimers can initiate a cycle through which they can remove actin from the JMY–actin complex, and afterward, the p53 can be trapped by GSN, and then JMY is able to collect actin monomers again (Figure 3A,B). In our interpretation, under polymerizing conditions, the continuous GSN-based severing and capping resulted in short actin filaments that are going to be truncated if JMY and p53 deplete monomers in the presence of millimolar Ca^2+^. Although, under nanomolar Ca^2+^ inactive GSN needed to bind actin monomers with a minor response to p53 binding to explain the response of actin to JMY addition, as JMY can collect actin filaments and monomers. Under micromolar Ca^2+^, actin filaments are capped by partially active GSN and then uncapped by subsequent p53 addition (Figure 3C), while JMY can bind both uncapped and capped filaments as well. Under millimolar Ca^2+^, active GSN continuously severs and caps filaments independently of p53, then JMY possibly binds to the pointed end of short filaments (Figure 3C). The JMY–actin complex shows a Ca^2+^-dependent response to p53 which was modified by the capping and/or severing activity of GSN (Figure 3D). Through comparison of Figure 2A,B and Figure 3B,D, we suggest that, in the presence of millimolar calcium and polymerizing salts (100 mM KCl, 1 mM MgCl_2_), JMY can deplete ATP–actin monomers, and p53 probably binds to actin filaments. If all ATP was hydrolyzed by actin, then the JMY affinity to actin monomers in turn be reduced, and therefore, p53 competes with ADP–actin monomer in JMY. There is a possible interpretation of Figure 3B that, in the absence of polymerizing salts, p53 can compete with ATP–actin monomer in JMY. We can presume that the p53 can bind simultaneously with actin monomer in GSN under nanomolar Ca^2+^.

Here we verified that a high level of cytoplasmic GSN can be localized nearby the nucleus (Figure 4H), as was shown in a previous study [11]. Our protocol for low-voltage electroporation stimulates HeLa cells, as they can move faster than non-electroporated cells in the first 5 h. High cytoplasmic JMY may translocated into the nucleus and resulted in the motility of cells slowing down [25,40,56,57], then they were divided faster than electroporated control cells. However, high cytoplasmic GSN after the electroporation resulted in cells moving normally and then dividing slowly (Figure 5A–F). Surprisingly, 24 h after the electroporation with JMY, the media was full of released cells (Figure 5H).

Based on these observations, we made a hypothetical model which is presented in the cartoon in Figure 6. Moderate stress caused by micromolar Ca^2+^ concentration activates the capping function of GSN, subsequently inducing continuous rearrangement of actin filaments. During short-term stress to the cell, the de novo filament synthesis can be stopped, and thus JMY can release from long-lasting filaments (mainly ADP–actin) and then can be transported into the nucleus. Based on our data, we suggest that an increased level of cytoplasmic p53 competes with actin on GSN which leads to uncapped filaments, thus GSN-trapped p53 inhibits p53–JMY complex formation and may prevent their transport into the nucleus. In the case of long-term stress, the millimolar cytoplasmic Ca^2+^ concentration induces the total activity of GSN and thus the severing and capping of filaments results in slow cell division while JMY sequestering actin monomers from the de novo filament synthesis results in low motility, as we observed. A high Ca^2+^ level initializes p53 dimerization, the dimer competes with actin on JMY and may help the p53–JMY cotransport into the nucleus to enhance p53 expression. Presumably, the sequestering of actin monomers and a high level of p53 by an unknown pathway may accelerate the cell division. We hypothesize that the reduced level of p53–JMY complex in the nucleus may prevent apoptosis, but, in contrast, expanded levels of p53–JMY complex expression may initialize apoptosis due to increased p53 expression.

## 4. Materials and Methods

### 4.1. Proteins

The human GSN (pSy5 construct, from Prof. Robert Robinson Lab, A*STAR IMCB, Singapore) and mouse JMY (pET20b construct, from Prof. Dyche Mullins Lab, UCSF, San Francisco, CA, USA) were expressed in *E. coli* BL21 Rosetta cells and purified with Ni-NTA affinity chromatography. They were then stored in Ca^2+^-free buffer A (0.2 mM ATP, 2 mM TRIS, pH 7.4). The purified human p53 was ordered directly from Sigma-Aldrich and then dialyzed against Ca^2+^-free storing buffer (2 mM TRIS, pH 7.4). Possible target residues of proteins are cysteine residues (p53: C124, 135, 141, 176, 182, 229, 238, 242, 275, 277; GSN: C93, 188, 201, 304, 645; JMY: C46, 208, 227, 246, 277, 291, 489, 538, 621, 708, 712, 788, 978) which were labeled with Alexa488–maleimide or Alexa568–maleimide (Life Technologies/Thermo Fisher, Waltham, MA, USA) with 40–50 % efficiency.

Rabbit skeletal muscle actin was prepared from acetone powder (Pel-Freez Biologicals/Gentaur USA, San Jose, CA, USA) using a method modified from Spudich and Watt [58,59]. Actin was stored in buffer A (2 mM Tris-HCl, 0.2 mM ATP, 0.1 mM CaCl_2_, 0.1 mM DTT and 0.005% NaN_3_, pH 7.4) and polymerized by 100 mM KCl, 1 mM MgCl_2_, 100 μM EGTA addition (means μM free Ca^2+^) supplemented with EGTA or CaCl_2_ to vary the free calcium levels (calculated with Maxchelator Stanford http://maxchelator.stanford.edu/CaMgATPEGTA-NIST.htm accessed several times in 2020 and 2021): for nM: 6 mM EGTA; for mM: 1 mM CaCl_2_.

### 4.2. Fluorescence Spectroscopy

We were using a Fluorolog-3 (Horiba, Jobin Yvon, Palaiseau, France) spectrofluorometer, typically with 3 nm slits, to measure spectra and time-dependent fluorescence emission with a 3 s integration time at 488 nm/516 nm and 568 nm/596 nm as the excitation/emission wavelength in the case of Alexa488 and Alexa568, respectively. The intrinsic tryptophan spectroscopy of human p53 was based on the residues of W23, W53, W91, and W146. They were excited at 290 nm and emitted at 330 nm in the absence and 332 nm in the presence of Ca^2+^. The fluorescence emission of N-(1-pyrenyl)iodoacetamide-labeled actin can be measured with 495 nm/516 nm as the excitation/emission wavelength. We determined steady-state fluorescence anisotropy using an average of 10 points with 5 s integration [45].

### 4.3. Cosedimentation

F-actin (2 μM) was incubated with JMY and/or p53 in buffer A completed with 6 mM EGTA or 1 mM CaCl_2_. Samples were centrifuged at 258,000× g for 30 min at room temperature. The pellets were resuspended and analyzed by SDS-PAGE. The band volumes were derived by ImageJ (was developed by NIH, Bethesda, MA, USA) and normalized with the sum of the pellet and supernatant bands.

### 4.4. Low-Voltage Electroporation

Previous studies published the benefits of low-voltage electroporation [52,53,54,55]. Based on our protocol, we applied an ArduinoUno^®^ board to a 5 V lead, giving a 2.3 mA current in 1 s ON/6 s OFF impulses for one hour to the cell culture to electroporate the proteins inside of the cells. HeLa cells were placed in their medium in a borosilicate-bottomed microscopic chamber, and then 2 days later, we refreshed the medium. Two holes were made in the lids to put in a one-one graphite electrode, subsequently connected to ArduinoUno. The medium contained the proteins that we wanted to get into the cells, typically in a 125 nM concentration. Images were taken 1, 2, 3, 4, 5, 12, 20, 22, and 24 h after the electroporation with an Olympus IX 71 light microscope (Olympus Europa Gmbh, Hamburg, Germany).

### 4.5. Cell Culture

HeLa cells were cultured in Alpha MEM (PAN Biotech, Aidenbach, Germany supplemented with 10% fetal bovine serum (Gibco/Thermo Fisher, Waltham, MA, USA). Cells were incubated in a humidified environment and 5% CO_2_ at 37 °C.

### 4.6. Cell Fixation, Labeling

After electroporation, cells recovered overnight, and then we conducted fixation and labeling with Alexa488–phalloidin with the following protocol. After removing the used medium, we washed the cells twice with PBS. For the fixation, cells were incubated in 4% paraformaldehyde for 15 min and were then washed in PBS. Afterward, cells were stored in a saturating-permeabilization solution (5% BSA, 0.1% TritonX, in PBS) for 30 min. For labeling the actin, the cells were incubated with Alexa488–phalloidin (0.3 μL Alexa488–phalloidin in 200 μL PBS) for 1 h. Then, after a final washing step in PBS, the samples were mounted and covered by a coverslip.

### 4.7. Microscopy, Confocal Microscopy

After fixation, HeLa cells were labeled by Alexa488–phalloidin. The two different fluorophores were detected in two separate channels; Alexa488 (phalloidin) was excited by 488 nm laser light and emission was detected in the 515–560 nm channel, while Alexa568 (GSN) was excited by 543 nm laser light and emission was detected in the 600–700 nm channel. We examined the cell morphology with Leica TCS SP confocal scanning microscope system (Leica Microsystems GmbH, Wetzlar, Germany) equipped with 10–63 × objective lenses. The most typical stacking size of images was 1–3 µm. For viable cell examinations, the Olympus IX 71 light microscope was applied with a 10 × objective lens.

### 4.8. Scratching Method

In the layer of HeLa cells, a roughly 400 μm wide, ‘T’-shaped gap was prepared by scratching with a sterile pipette tip. The rate of cell division and motility was measured by the change in the gap size after treatment at 5, 12, and 24 h (the generation time of HeLa cells is 16–28 h [60,61,62]). The normalized change of the gap size gives information about the cellular motility in the first 5 h and the division rate at 12 and 24 h. The time-dependent recovery of the scratch depends on the initial size of the gap and the size of the cells. These parameters were used to normalize the change in the gap size. For the analysis of the cellular motility and division rates in cells/hour, we used the normalized values in the first 5 h and 24 h, respectively. The cell motility was visualized with an Olympus IX 71 microscope.

### 4.9. Image Analysis

ImageJ (was developed by NIH, Bethesda, MA, USA) was applied for the quantitative analysis of microscopic images.

### 4.10. Statistics

The data presented were derived from at least 3 independent experiments as means ± standard deviations (SD) throughout. Comparisons were performed using Student’s *t*-test, and statistically significant differences between groups were defined as *p* values < 0.05 and are indicated in the legends of the figures.

## 5. Conclusions

Our data suggest that Ca^2+^ concentration-dependent p53 complexes with GSN or JMY can play an important role in the permanent stress response of cells and possibly help JMY nuclear translocation and increase free p53 concentration in the cytoplasm, regulating the quick rearrangement of filamentous actin and the motility of cells. Moderate stress-caused cytoskeletal remodeling can be coupled to the process of p53 and Ca^2+^-dependent uncapping of actin filaments, and cytoplasmic accumulation of JMY. However, the cytoplasmic concentration of GSN and JMY may are alternative signals to a mechanism which could be linked to cell cycle control.

## Figures and Tables

**Figure 1 ijms-23-09078-f001:**
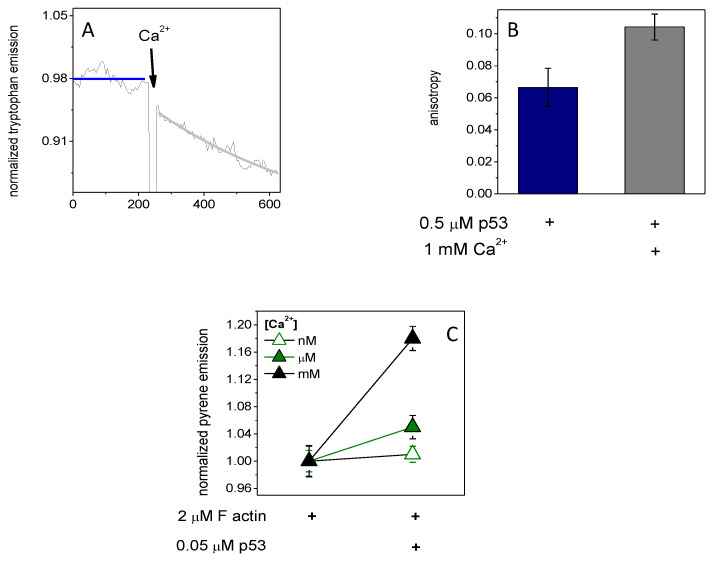
The intrinsic tryptophan fluorescence emission and anisotropy of p53 tumor suppressor factors were changed by Ca^2+^ addition, and Ca^2+^ helps p53 binding to actin. (**A**) The intrinsic tryptophan fluorescence emission of p53 (0.5 μM) was decreased by millimolar Ca^2+^ addition and (**B**) the steady-state anisotropy was increased. (**C**) The fluorescence emission of pyrene-labeled actin (2 μM, 5% labeled by pyrene) under polymerizing conditions (100 mM KCl, 1 mM MgCl_2_) in the presence of micromolar (green, closed triangle) then millimolar (black, closed triangle) Ca^2+^ was increased by p53 (0.05 μM), which could not be observed in the presence of nanomolar (open green triangle) Ca^2+^. The data presented were derived from at least three independent experiments. Values are displayed as the mean ± standard deviation.

**Figure 2 ijms-23-09078-f002:**
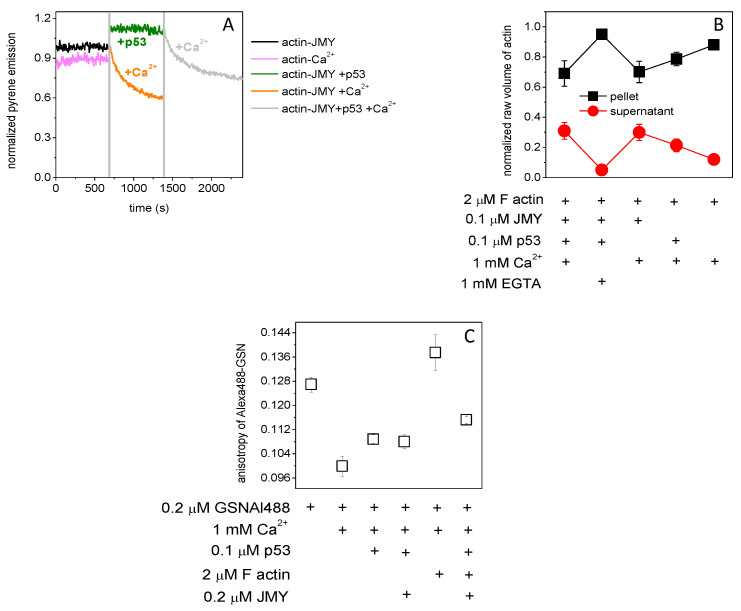
JMY reorders actin filaments. (**A**) The pyrene–actin (2 μM, 5% labeled by pyrene) was polymerized (by 0.1 mM EGTA, 100 mM KCl, 1 mM MgCl_2_) in the presence of JMY (0.1 μM) (black line). The pyrene emission was increased by p53 (0.1 μM) (green line) and then subsequently decreased by 1 mM Ca^2+^ addition (gray line). In the absence of p53, JMY–actin shows a bit higher sensitivity to millimolar Ca^2+^ (orange line), while in the absence of JMY and p53, it was less sensitive to Ca^2+^ (pink line). (**B**) The analysis of filamentous actin co-sedimentation (2 μM) with p53 and/or JMY (by SDS-PAGE Suppl. 1A) shows a big loss of actin which went from the pellet to the supernatant in the presence of 1 mM Ca^2+^ and 0.1 μM JMY, independently of p53. (**C**) The fluorescence anisotropy of Alexa488-labeled GSN (0.2 μM) under polymerizing conditions (100 mM KCl, 1 mM MgCl_2_) in the presence of millimolar Ca^2+^ with p53 (0.1 μM) was not changed by JMY (0.2 μM). The anisotropy was increased by 28% by filamentous actin (2 μM) in the absence of p53 and increased by only 8% in the presence of p53, JMY and filamentous actin. The data presented were derived from at least three independent experiments. Values are displayed as the mean ± standard deviation.

**Figure 3 ijms-23-09078-f003:**
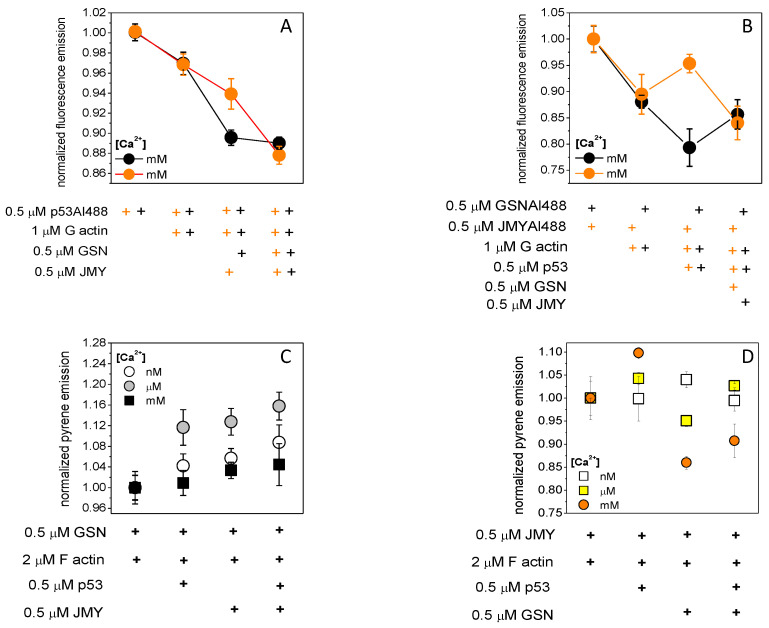
Actin competes with p53 on JMY and p53 competes with actin on GSN in a calcium-dependent manner. (**A**) The emission of 0.5 μM p53Alexa488 was reduced by 3% by 1 μM actin then 3% or 10% more by subsequently adding 0.5 μM JMY (orange circles) or 0.5 μM GSN (black circles), respectively, in the absence of polymerizing salts (100 mM KCl, 1 mM MgCl_2_) and in the presence of 1 mM Ca^2+^. The emission was not changed by the addition of JMY in the presence of actin and GSN, but it was reduced by 7% by GSN in the presence of actin and JMY (orange plus belongs to orange circle, black plus belongs to black circle). (**B**) The fluorescence emission of 0.5 μM JMYAlexa488 (orange circles) was reduced by 11% by 1 μM actin then increased by 5% by subsequently adding 0.5 μM p53 in the absence of polymerizing salts and in the presence of 1 mM Ca^2+^. The fluorescence emission of 0.5 μM GSNAlexa488 (black circles) was reduced by 13% by actin then decreased 10% more by subsequently adding p53. The emission of JMYAlexa488 was reduced by 12% by 0.5 μM GSN in the presence of actin and p53; in addition, the emission of GSNAlexa488 was increased by 8% by 0.5 μM JMY in the presence of actin and p53. (**C**) Pyrene–actin (2 μM, 5% labeled by pyrene) was polymerized (100 mM KCl, 1 mM MgCl_2_) in the presence of 0.5 μM GSN. The fluorescence emission of pyrene was increased by 0.5 μM p53 then subsequent 0.5 μM JMY addition in the presence of nanomolar (4–8%, open circles) than micromolar (12–16%, grey circles) and millimolar (4%, closed black circles) Ca^2+^. The emission of pyrene–actin was increased even by JMY addition under micromolar (grey circles) Ca^2+^. It was increased by 12% and 3 and 6% under nanomolar (open circles) and millimolar (closed circles), respectively. (**D**) Pyrene–actin was polymerized in the presence of 0.5 μM JMY; the emission was increased by 0.5 μM p53 addition in a Ca^2+^ concentration-dependent manner as under nanomolar 0% (open squares), under micromolar 4% (yellow squares), and under millimolar 10% (orange circles) increment, then the subsequent addition 0.5 μM GSN reduced the pyrene emission by 2% under micromolar and by 20% under millimolar Ca^2+^. In the absence of p53, the emission of JMY-bound pyrene–actin was increased by 4% under nanomolar Ca^2+^ (open squares) and decreased by 5% and 15% under micromolar (yellow squares) and millimolar (orange circles) Ca^2+^, respectively. The data presented here were derived from at least three independent experiments. Values are displayed as the mean ± standard deviation.

**Figure 4 ijms-23-09078-f004:**
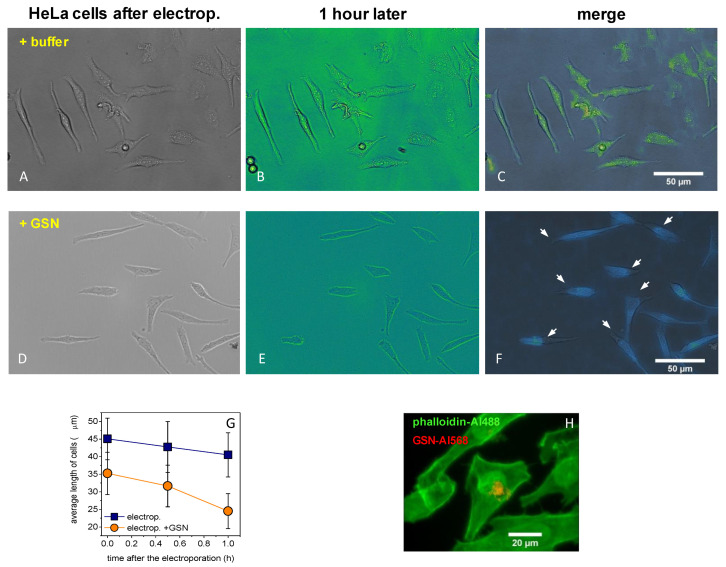
Increased cytoplasmic GSN level in HeLa cells. We carried out low-voltage electroporation on HeLa cells to increase the GSN concentration in the cytoplasm. (**A**–**C**) The cell size did not show any difference in the first hour after the electroporation with buffer only (**D**–**F**), but it was reduced by electroporation with 125 nM GSN (white arrows indicate the positions of changes). (**G**) The average cell length was significantly decreased by the applied GSN electroporation (*p* < 0.05). (**H**) HeLa cells were successfully electroporated with GSN–Alexa568 (red), and then samples were fixed and labeled with phalloidin–Alexa488 (green). The data presented here were derived from at least three independent experiments. Values are displayed as the mean ± standard deviation.

**Figure 5 ijms-23-09078-f005:**
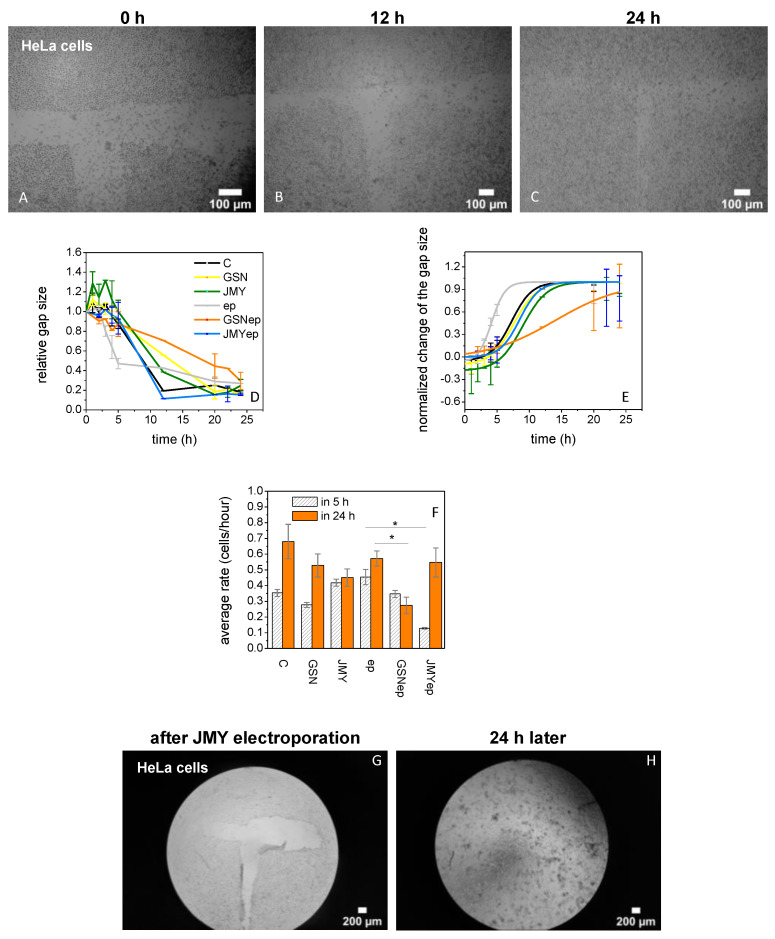
GSN and JMY affect the division and motility of HeLa cells. (**A**–**C**) Scratching assays to examine the rate of gap size recovery after GSN or JMY electroporation. In the first 24 h after scratching a ’T’ shape in HeLa cells, the cultures were examined to investigate the effect of GSN or JMY electroporation. (**D**) The assays provided data about the effect on the change of the relative gap size by simple addition of buffer only (black line), 125 nM GSN (yellow line), 125 nM JMY (green line) to the cell culture or electroporation with buffer only (gray line), with 125 nM GSN (orange line), or with 125 nM JMY (blue line). (**E**) The normalized change of the gap size gives information about the cellular motility changes in the first 5 h and the division rate at 12 and 24 h. The time-dependent recovery of the scratch is related to the initial size of the gap and the size of the cells. These parameters were used to normalize previous data. The electroporation with GSN shows a different slope of the sigmoid graph. (**F**) For the analysis of the cellular motility and division rates in cells/hour, we used the normalized values in the first 5 h and 24 h, respectively. The comparison of recovery rates resulted that the electroporation stimulated the cells which were moving faster than non-electroporated cells in the first 5 h. After the electroporation with JMY, the motility of cells was slowed down (* *p* < 0.05), after they divided faster than electroporated control cells. Due to the electroporation with GSN, cells were moving normally but dividing slowly (*p* < 0.05). (**G**,**H**) HeLa cells recovered the gap fully 20 h after the electroporation with JMY then started to release from the glass slides to the solution. The data presented were derived from at least three independent experiments. Values are displayed as the mean ± standard deviation.

**Figure 6 ijms-23-09078-f006:**
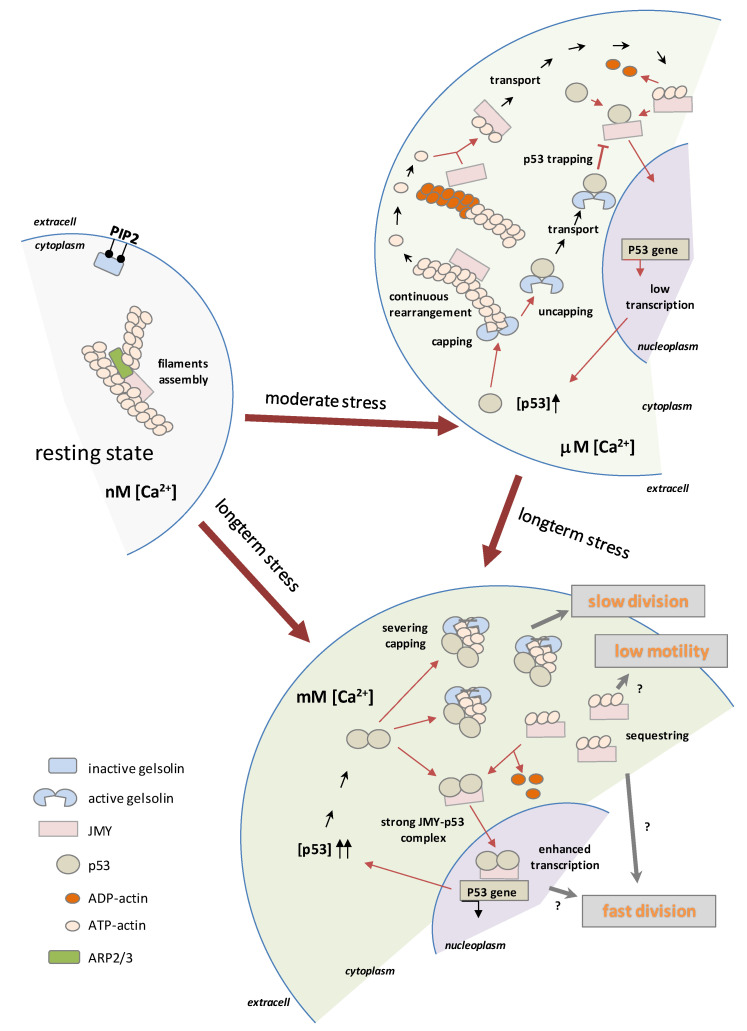
Hypothetical model of p53- and Ca^2+^-regulated function of GSN and JMY. The moderate stress caused by the micromolar Ca^2+^ concentration activates the capping function of GSN and subsequently induces the continuous rearrangement of actin filaments. During short-term stress to the cell, de novo filament synthesis can be stopped, and thus JMY can be released from long-lasting filaments (mainly ADP–actin) and then transported into the nucleus. Increased levels of cytoplasmic p53 compete with actin in GSN and lead to uncapped filaments, thus trapped p53 cannot form complexes with JMY, and due to this, may prevent the nuclear transport of p53–JMY and cannot enhance the expression of p53. In the case of a long-term stress, the millimolar Ca^2+^ concentration induces the total activity of GSN, and thus, through the severing and capping of filaments, results a slow division of cells, all while the JMY sequestering of actin monomers takes them out of the de novo filament synthesis and results in slow cell motility. A high Ca^2+^ level initializes p53 dimerization, and the dimer competes with actin in JMY which helps p53–JMY cotransport into the nucleus, thereby initializing enhanced p53 expression. (symbols of “?” indicate the unknown relationships).

## Data Availability

The manuscript contains all the data which were generated within this study.

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
