# Peer review of "The p53 and Calcium Regulated Actin Rearrangement in Model Cells"

_ijms, 2022, doi:10.3390/ijms23169078_

Round 1
Reviewer 1 Report
In this manuscript, the authors investigated the synergistical effects between p53, JMY and gelsolin on actin cytoskeleton dynamics using in vitro fluorescence spectroscopy and wound healing assay with HeLa cells electroporated with gelsolin or JMY. The work is of good novelty and scientific significance, and is highly relevant to the topic of calcium signaling and calcium transport in tumors. However, the data in this manuscript need to be presented and discussed in a clearer manner before the manuscript can be published. Below are some suggestions for the authors to consider.
In Figure 1A, the authors need to explain the reason for the steady decline of the signal before Ca2+ addition, and the red shift the the spectra doesn’t seem to be convincing. The authors may also consider to rephrase the title of the caption to better reflect the contents of the figure.
Figure S1 is missing for review, so I can’t make any comments on it.
In Figure 2A, the authors may added the response of actin alone (described on Page 3 as “in the absence of JMY and p53 it was only 7 %.”) to highlight the changes caused by JMY and p53. In Figure 2B, the EGTA data points seem to be unnecessary. In Figure 2C, the authors should add data without p53 and comment if the anisotropy measurement can readily detect gelsolin-p53 interaction.
In Figure 3, C & D could be merged into one, and so do E & F. For all such figures, common components should be omitted from the panel below the figure to highlight the differences between each column.
For the Discussion, the authors should refer to the Figures, so as to establish clearer links between the statements they make and the experimental results.
For the Conclusion and Figure 6, the authors should clearly distinguish between steps with experimental evidence and those without. For example, uncapping by p53 and complex formation between monomeric p53 and JMY don’t seem to have any experimental data support.
In summary, to improve the readability of the manuscript, better presentation and discussion of the available data are expected.
Author Response
Summary: We thank the Editorial Board Members and Reviewers for their helpful and constructive comments. Additions or modifications made in direct responses are shown below. Changes made in response to those comments.
Reviewer Comments:
Answers to Reviewer 1.
In Figure 1A, the authors need to explain the reason for the steady decline of the signal before Ca2+ addition, and the red shift the the spectra doesn’t seem to be convincing.
Thank you, for these observations. We made exponential fitting onto both segments and have got 100 times lower decay constant with the noisy segment before than after the Ca2+ addition. The p53 is a sensitive protein need to keep carefully and somehow show some minor fluctuations (± 2 %) of the tryptophan emission. But we agree your concern as difficult to interpret the decreases signal thus we made a linear fitting onto the first segment and got a nearly constant line.
We agree the negligible importance of small spectral shifts thus we removed the inner panel of Figure 1A.
The authors may also consider to rephrase the title of the caption to better reflect the contents of the figure.
Thank you for this important recommendation, we modified it.
Figure S1 is missing for review, so I can’t make any comments on it.
I am so sorry, probably the link on the site to the supplements was not enoughly remarkable. I will let administrators know about this issue.
In Figure 2A, the authors may added the response of actin alone (described on Page 3 as “in the absence of JMY and p53 it was only 7 %.”) to highlight the changes caused by JMY and p53.
Thank you for this important observation. The data mentioned above is already on the figure, it was labeled with pink color. It was our mistake that did not describe the differences between the different colors thus we did a make up in Figure legends.
In Figure 2B, the EGTA data points seem to be unnecessary.
To reflect to the aboves, I think the data points with EGTA verifies the importance of Ca2+ addition, as the biggest difference between pellet and supernatant was generated by the EGTA treatment caused nanomolar free Ca2+, in the presence of p53 and JMY. Interestingly, in the absence of JMY and p53 but in the presence of milimolar Ca2+ the pellet size was almost the same as in the presence of JMY, p53 and nanomolar Ca2+.
In Figure 2C, the authors should add data without p53 and comment if the anisotropy measurement can readily detect gelsolin-p53 interaction.
Thank you for this important recommendation, we measured it but did not think it was neccessary to publish again anisotropy data based on calcium induced structrual changes of GSN. Refers to previous studies (e.g. Szatmári et al. 2018 PlosOne[1], Pintér et al. 2020 Front.Mol.Biosci.[2]), anisotropy measurement can show direct interaction between proteins. We completed the Figure 2C with more data. And it was briefly commented in Part of Results as „The steady-state anisotropy change of GSN-Al488, indicates slowing thermal rotational motion, reveals Ca2+ caused structural changes, and the p53 binding, severing and capping activity of GSN on short actin filaments (under polymerizing conditions 100 mM KCl, 1 mM MgCl2) in the presence of millimolar Ca2+ with JMY and/or p53 (Figure 2C).”
In Figure 3, C & D could be merged into one, and so do E & F. For all such figures, common components should be omitted from the panel below the figure to highlight the differences between each column.
Thanks to the Reviewer, it made that differences more impressed.
For the Discussion, the authors should refer to the Figures, so as to establish clearer links between the statements they make and the experimental results.
Thank you. We linked Figures to statements.
For the Conclusion and Figure 6, the authors should clearly distinguish between steps with experimental evidence and those without. For example, uncapping by p53 and complex formation between monomeric p53 and JMY don’t seem to have any experimental data support.
Thanks to the Reviewer for this important suggestion. Thus, we have corrected these sentences and moved it in the Part of Discussion as „Based on these observations we made a hypothetical model is presented in the cartoon in Figure 6, as moderate stress caused by micromolar Ca2+ concentration activates the capping function of GSN, subsequently inducing continuous rearrangement of actin filaments. During short stress of cell, the de novo filament synthesis can be stopped thus JMY can release from long-lasting filaments (mainly ADP-actin) and then can be transported into the nucleus. Based on our data we suggest that an increased level of cytoplasmic p53 competes with actin on GSN which leads to uncapped filaments, thus GSN trapped p53 inhibits p53-JMY complex formation and may prevent their transport into the nucleus. In case of long-term stress, the millimolar cytoplasmic Ca2+ concentration induces the total activity of GSN and thus severing and capping of filaments results in a slow cell division while the JMY sequestering actin monomers from the de novo filament synthesis results in a low motility, as we observed. A high Ca2+ level initializes p53 dimerization, the dimer competes with actin on JMY may help the p53-JMY cotransport into the nucleus to enhance p53 expression. Presumably, sequestering of actin monomers and high level of p53 by an unknown pathway may accelerate the cell divison. We hypothetise that the reduced level of p53-JMY complex in the nucleus may prevent apoptosis, but in contrast expanded level of p53-JMY complex expression may initialize apoptosis due to increased p53 expression.”
In summary, to improve the readability of the manuscript, better presentation and discussion of the available data are expected.
Thank you for your important and constructive comments, we tried to improve the quality of our manuscript.
- Szatmari, D., et al., ATP competes with PIP2 for binding to gelsolin. Plos One, 2018. 13(8).
- Pinter, R., et al., The Activities of the Gelsolin Homology Domains of Flightless-I in Actin Dynamics. Front Mol Biosci, 2020. 7: p. 575077.

Reviewer 2 Report
Overview: This manuscript by Hencz et al investigated the effects of the tumor protein (p53) on actin, actin-gelsolin (GSN) and actin-Y protein (JMY) complex interactions and its link to apoptotic pathways due to stress responses. These two actin binding proteins were chosen because of their significance in cancer formation–moreover p53 and GSN/JMY are affected by cytoplasmic Ca2+ concentration increase which is induced by stress response.
Because GSN/JMY actin binding proteins are present in short-term and long-term stress response, the authors aimed to answer two main questions: 1) is there a direct relationship to malignant cytoskeletal transformation and 2) how can GSN/JMY be linked to cytoskeletal remodeling and motility alteration in the apoptotic pathway.
They studied p53 interactions with actin and the aforementioned complexes under different Ca conditions, along with the use of HeLa cells and electroporations to observe the cell motility and divisions rate in the presence of GSN or JMY.
The manuscript can be improved by addressing comments below:
1) Introduction and Abstract could be written more clearly and concisely to present important background information and main research problem. Also, I find many sentences within this paper are riddled with English grammar mistakes. The manuscript should be proofread more thoroughly.
2) It is not clear how p53 factor forms dimers based on their data (Figure1). Can this part be elaborated in Section 2.1? In addition, it has been stated that p53 competes with actin on GSN and also in high Ca2+ levels p53 forms dimers which competes with actin for JMY. What is the probability of different complexes forming in the cell?
3) For the fluorescence and pyrene measurements displayed in Figure 1c and Figure 3, What concentrations were the Calcium at for each of the conditions? how long was the actin incubated? Did the authors wait for the actin to reach steady state?
4) For co-sedimentation results in Figure 2b, there is no obvious difference in Actin+JMY+Ca2+ samples wit the inclusion (1st column) or absence (3rd column) of p53. What does this mean in regard to JMY’s competitive binding with p53 and actin?
5) For anisotropy measurements in Figure 2c, it is stated: “It seems that the JMY or p53 inhibits the GSN binding to actin in the presence of milimolar Ca2+” why not perform measurements of GSN + Actin + Ca2+ + either GSN or p53 alone to differentiate which of the two had the greater effect after addition of the 2μM F-actin?
6) For their in vivo studies, how similar are Hela cells apoptotic signaling pathways, gene expression levels, and motility rates compared to regular cells? As Hela cells are already cancerous and should have some measures in place to prevent apoptosis and have different protein expression levels, wouldn’t it be more appropriate to use a non-cancerous cell line to study the effects of stress on cells and how the subsequent signaling would induce malignant cytoskeletal rearrangements or apoptosis.
7) I suggest placing the mechanism figure (Figure 6) in Discussion rather than in Conclusions.
Author Response
Summary: We thank the Editorial Board Members and Reviewers for their helpful and constructive comments. Additions or modifications made in direct responses are shown below. Changes made in response to those comments.
Answers to Reviewer 2.
- Introduction and Abstract could be written more clearly and concisely to present important background information and main research problem. Also, I find many sentences within this paper are riddled with English grammar mistakes. The manuscript should be proofread more thoroughly.
Thank you for this important recommendation, we corrected and made a more thorough proofread.
- It is not clear how p53 factor forms dimers based on their data (Figure1). Can this part be elaborated in Section 2.1?
Thank you for this important question. Previous studies (Banik et al. 1993[1], Zorilla et al. 2004[2], Koker et. al. 2018[3]) interpreted the advantages of anisotropy measurement to explain the dimerization caused size increment of proteins. To reflect to the aboves, we included the following sentences in the Part of 2.1 : „The decreasing emission can be interpreted as structural change of p53 which resulting a burried state of tryptophan residues thus their fluorescence emission was decreased. The increased anisotropy can be described by slower thermal rotational motion of the protein. If tryptophan residues were turned to be burried in the structure of p53 which combined with a slower rotational motion of the whole protein implies a process of structural stabilization with a size increment, this obsrevation can be interpreted as a calcium-induced dimerization of p53. “
In addition, it has been stated that p53 competes with actin on GSN and also in high Ca2+ levels p53 forms dimers which competes with actin for JMY. What is the probability of different complexes forming in the cell?
Thanks to the Reviewer for this interesting question. An et al. 2011[4] published a study that GSN can bind p53 and their complex was localized nearby of the nuclear envelop. There is no direct data to explain p53-JMY complexes in cytoplasm, however we can be sure that p53 forms several different type of complexes [5] those can be regulated by posttranslational modifications[6]. Baptiste et al. 2004 [7] published the cytoplasmic transactivator role of p53 and thus verifed p53 binding to other proteins. Adighibe et al. 2014 [8] suggested that p53 expression shows a positive correlation with cytoplamsic and nuclear localization of JMY, in 2018[9] in a review summarized that JMY actin monomer interaction stops JMY entry into the nucleus, thereafter by an unknown import process JMY has to bind p53 tetramer in the nucleoplasm. In our model, we try to predict the steps related to actin release from JMY in cytoplasm for the entry of JMY into the nucleus followed by a subsequent p53tetramer/p300/STRAP binding.
- For the fluorescence and pyrene measurements displayed in Figure 1c and Figure 3, What concentrations were the Calcium at for each of the conditions?
Thanks to the Reviewer for this important point, in Figure 1C open triangle means nanomolar, green triangle means micromolar, closed triangle means milimolar. In Figure 3A,B the calcium concentration was 1 mM. In Figure 3 C open circle means nanomolar, gray circle means micromolar, closed circle means millimolar Ca2+. In Figure 3 D open square means nanomolar, yellow square means micromolar, orange circle means millimolar Ca2+.
how long was the actin incubated? Did the authors wait for the actin to reach steady state?
Before the experiments all complexes with actin ( p53Al488-actin, GSNAl488-actin, JMYAl488-actin, GSN-F actin, JMY-F actin) were incubated overnight, on ice in a fridge at 4 °C. We were waiting until the stabilization of steady-state fluorescence emission what was neccessary for the analysis.
- For co-sedimentation results in Figure 2b, there is no obvious difference in Actin+JMY+Ca2+samples wit the inclusion (1st column) or absence (3rd column) of p53. What does this mean in regard to JMY’s competitive binding with p53 and actin?
Thank you for this interesting and important question. The complex formation between actin, JMY, and p53 is not completely understood yet. Saha et al. 2016 [10] described that p53 can bind actin monomer, Metcalfe et al.[11] already in 1999 published that p53 binds filamental actin in the presence of millimolar calcium. Mattila et al. 2003 [12] published that the binding of three WH2 (Wiskott–Aldrich syndrome protein homology 2 domains) domains to actin depends on the bound nucleotide type, therefore it shows strong interaction with ATP-actin (Kd = 60 nM) and lower affinity to ADP-actin (Kd = 300 nM) under polymerizing conditions (0.1 mM CaCl2, 100 mM KCl, 1 mM MgCl2). We suggest that JMY can prefer monomeric ATP-actin and show less affinity to ADP-actin within filaments thus can do sequestering of monomers and induce depolymerization of filaments. Direct interaction of p53 with actin also can favor monomer release from filaments. Refers to Fig.S1 D, JMY can bind p53 only in the presence of millimolar calcium, but calcium helps the p53-actin interaction. Based on Figure2 A,B and Figure3 B,D we can suggest that in the presence of millimolar calcium and polymerizing salts (100 mM KCl, 1 mM MgCl2) JMY can deplet ATP-actin monomers and probably p53 binds to actin filaments. If all ATP was hydrolyzed by actin, the JMY affinity to actin monomers was turned to be reduced therefore p53 competes with ADP-actin on JMY. There is a possible interpretation of Figure3 B as in the absence of polymerizing salts p53 can compete with ATP-actin monomer on JMY.
To reflect the aboves, we completed the Part of Discussion with: „The JMY-actin complex shows a Ca2+ dependent response to p53 which was modified by capping and/or severing activity of GSN (Fig.3 D). By the comparison of Figure2 A,B and Figure3 B,D we suggest that in the presence of milimolar calcium and polymerizing salts (100 mM KCl, 1 mM MgCl2) JMY can deplet ATP-actin monomers and probably p53 binds to actin filaments. If all ATP was hydrolyzed by actin, the JMY affinity to actin monomers was turned to be reduced therefore p53 competes with ADP-actin monomer on JMY. There is a possible interpretation of Figure3 B, that in the absence of polymerizing salts, p53 can compete with ATP-actin monomer on JMY.”
- For anisotropy measurements in Figure 2c, it is stated: “It seems that the JMY or p53 inhibits the GSN binding to actin in the presence of milimolar Ca2+” why not perform measurements of GSN + Actin + Ca2+ + either GSN or p53 alone to differentiate which of the two had the greater effect after addition of the 2μM F-actin?
Thanks to the Reviewer for this important observation, our statement can have a misleading function. Refers to Figure 3 and the above expressed concern of the Reviewer in case of JMY, p53, and actin complex formation, we can interpret Fig2 C as GSN need free surface on actin filaments to do severing and capping but if the length and number of filaments were reduced by JMY and p53 the GSN binds to remaining oligomers and monomers leads to low increment of anisotropy. We made a modification in section 2.2 as „It seems that the JMY and p53 can reduce the length and number of filaments and thus GSN can bind only short oligomers and monomers which was manifested in a small change of anisotropyin the presence of millimolar Ca2+.”
- For their in vivo studies, how similar are Hela cells apoptotic signaling pathways, gene expression levels, and motility rates compared to regular cells? As Hela cells are already cancerous and should have some measures in place to prevent apoptosis and have different protein expression levels, wouldn’t it be more appropriate to use a non-cancerous cell line to study the effects of stress on cells and how the subsequent signaling would induce malignant cytoskeletal rearrangements or apoptosis.
Thank you for this important question. It was a misleading choice of us to use the concept of apoptosis, first in the title then in the rest of the manuscript. Our observations and data with in vitro cells are related to HeLa as model cells rather than to apoptotic cells. We had to reconsider and modify the concept and statements in the rest of the mansucript.
- I suggest placing the mechanism figure (Figure 6) in Discussion rather than in Conclusions.
Thank you for this recommendation. We moved it in the Part of Discussion.
Thank you for your important and constructive comments, we tried to improve the quality of our manuscript.
- Banik, U., et al., A fluorescence anisotropy study of tetramer-dimer equilibrium of lambda repressor and its implication for function. J Biol Chem, 1993. 268(6): p. 3938-43.
- Zorrilla, S., et al., Protein self-association in crowded protein solutions: A time-resolved fluorescence polarization study. Protein Science, 2004. 13(11): p. 2960-2969.
- Koker, T., A. Fernandez, and F. Pinaud, Characterization of Split Fluorescent Protein Variants and Quantitative Analyses of Their Self-Assembly Process. Scientific Reports, 2018. 8.
- An, J.H., et al., Gelsolin negatively regulates the activity of tumor suppressor p53 through their physical interaction in hepatocarcinoma HepG2 cells. Biochemical and Biophysical Research Communications, 2011. 412(1): p. 44-49.
- Cao, H., et al., The role of MDM2-p53 axis dysfunction in the hepatocellular carcinoma transformation. Cell Death Discov, 2020. 6: p. 53.
- Lavin, M.F. and N. Gueven, The complexity of p53 stabilization and activation. Cell Death Differ, 2006. 13(6): p. 941-50.
- Baptiste, N. and C. Prives, p53 in the cytoplasm: a question of overkill? Cell, 2004. 116(4): p. 487-9.
- Adighibe, O., et al., JMY protein, a regulator of P53 and cytoplasmic actin filaments, is expressed in normal and neoplastic tissues. Virchows Arch, 2014. 465(6): p. 715-22.
- Adighibe, O. and F. Pezzella, The Role of JMY in p53 Regulation. Cancers (Basel), 2018. 10(6).
- Saha, T., et al., G-actin guides p53 nuclear transport: potential contribution of monomeric actin in altered localization of mutant p53. Sci Rep, 2016. 6: p. 32626.
- Metcalfe, S., et al., Wild-type p53 protein shows calcium-dependent binding to F-actin. Oncogene, 1999. 18(14): p. 2351-5.
- Mattila, P.K., et al., Mouse MIM, a tissue-specific regulator of cytoskeletal dynamics, interacts with ATP-actin monomers through its C-terminal WH2 domain. J Biol Chem, 2003. 278(10): p. 8452-9.

Reviewer 3 Report
The manuscript entitled “The p53 and calcium regulated actin rearrangement in apoptotic cells“ by Alexandra Hencz et al. reports a study of the regulation of apoptosis and cell fate by the Ca2+ concentration which determines the complex formation between p53 and GSN/JMY and subsequently the rearrangement of actin filaments.
There are few comments I would like the authors to address:
1. p53 is active in a form of a tetramer. Here, the authors claim that in the form of dimers p53 binds to actin. Are you sure about that?
2. I would avoid the expression “apoptotic factor” for p53.
3. To confirm the idea that the cells enter the apoptosis or not, it should be tested.
4. The English should be improved.
Author Response
Summary: We thank the Editorial Board Members and Reviewers for their helpful and constructive comments. Additions or modifications made in direct responses are shown below. Changes made in response to those comments.
Reviewer Comments:
Answers to Reviewer 3.
- p53 is active in a form of a tetramer. Here, the authors claim that in the form of dimers p53 binds to actin. Are you sure about that?
Thank you for this important question. Previous studies (Banik et al. 1993[1], Zorilla et al. 2004[2], Koker et. al. 2018[3]) interpreted the advantages of anisotropy measurement to explain the dimerization caused size increment of proteins. The intrinsic tryptophan fluorescence emission of p53 was decreased due to increases Ca2+ concentration. It was observable along with an increased steady-state anisotropy. The decreases emission can be interpreted as structural change of p53 which resulted a burried state of tryptophan residues thus their fluorescence emission was decreased. The increases fluorescence anisotropy can be described by slower thermal rotational motion of the whole protein. If tryptophan residues were turned to be burried in the structure of p53, and even combined with a slower rotational motion of the whole protein implies a process of structural stabilization with size increment, this observation can be interpreted as a calcium induced dimerization of p53. The tetramer can be produced by the dimerization of dimers and it happens most typically in the nuceloplasm as required step for high affinity DNA binding [4]. We did not do any analytical assay to figure out stoichiometry of p53 –p53 interactions, but the fluorescent anisotropy change need to show different kinetics with dimer then tetramer formation if it is measurable. Therefore we can not be sure do we have p53 dimers or tetramers but as structural studies predicted [5, 6] dimer formation is refining the most stable structure of p53. Several studies simply explain the cytoplasmic function of p53 as do it in monomer form [7] and lack of function type of mutations helps the cytoplasmic depletion of dimers [8, 9] seems that dimers are more frequent in cytoplasm and for the nucleoplasmic function of p53 need to form tetramers. Rajagopalan et al. 2008 [10] published the reason that p53 phophorylation was needed for the stability of tetramers. Although, in FigureS1 B-E data were fitted with hyperbole equation to determine Kd values of p53-GSN and p53-JMY, we are able to modify it with the stoichiometry of complexes, if we use 4 : 1, as p53 : GSN or p53 : JMY, the Kd value increases definitely. Finally, we decided to use data based on dimers to interpret them with high affinities.
- I would avoid the expression “apoptotic factor” for p53.
Thank you for this important observation, we corrected them.
- To confirm the idea that the cells enter the apoptosis or not, it should be tested.
Thank you for this important question. It was a misleading choice of us to use the concept of apoptosis, first in the title then in the rest of the manuscript. Our observations and data with in vitro cells are related to HeLa as model cells rather than to apoptotic cells. We had to reconsider and modify the concept and statements in the rest of the mansucript.
- The English should be improved.
Thank you for your important and constructive comments, we tried to improve the quality of our manuscript.
- Banik, U., et al., A fluorescence anisotropy study of tetramer-dimer equilibrium of lambda repressor and its implication for function. J Biol Chem, 1993. 268(6): p. 3938-43.
- Zorrilla, S., et al., Protein self-association in crowded protein solutions: A time-resolved fluorescence polarization study. Protein Science, 2004. 13(11): p. 2960-2969.
- Koker, T., A. Fernandez, and F. Pinaud, Characterization of Split Fluorescent Protein Variants and Quantitative Analyses of Their Self-Assembly Process. Scientific Reports, 2018. 8.
- Chene, P., The role of tetramerization in p53 function. Oncogene, 2001. 20(21): p. 2611-7.
- Ho, W.C., M.X. Fitzgerald, and R. Marmorstein, Structure of the p53 core domain dimer bound to DNA. J Biol Chem, 2006. 281(29): p. 20494-502.
- Madhumalar, A., et al., Dimerization of the core domain of the p53 family: a computational study. Cell Cycle, 2009. 8(1): p. 137-48.
- Green, D.R. and G. Kroemer, Cytoplasmic functions of the tumour suppressor p53. Nature, 2009. 458(7242): p. 1127-30.
- Baptiste, N. and C. Prives, p53 in the cytoplasm: a question of overkill? Cell, 2004. 116(4): p. 487-9.
- Morselli, E., et al., Mutant p53 protein localized in the cytoplasm inhibits autophagy. Cell Cycle, 2008. 7(19): p. 3056-61.
- Rajagopalan, S., et al., 14-3-3 activation of DNA binding of p53 by enhancing its association into tetramers. Nucleic Acids Res, 2008. 36(18): p. 5983-91.

Round 2
Reviewer 2 Report
The authors addressed most of my comments in their revised manuscript satisfactorily.
A few minor changes are recommended for clarity:
1. Label F-actin or F actin instead of "Factin" in some figures to avoid any confusion. (e.g. Figure 1C, Figure 2B)
2. Figure 5 graphs: Make sure axis labels are legible in a printed form.
3. Conclusion paragraph may be revised to summarize key points of the paper better.
Author Response
Summary: We thank the Editorial Board Members and Reviewers for their helpful and constructive comments. Additions or modifications made in direct responses are shown below. Changes made in response to those comments.
Reviewer Comments:
Answers to Reviewer 2.
- Label F-actin or F actin instead of "Factin" in some figures to avoid any confusion. (e.g. Figure 1C, Figure 2B)
Thank you, for these observations. We made a correction with above mentioned Figures and checked it in the text.
- Figure 5 graphs: Make sure axis labels are legible in a printed form.
Thank you, for this suggestion. We improved their quality.
- Conclusion paragraph may be revised to summarize key points of the paper better.
Thank you, we agree the importance of the Part of Conclusion thus we tried to improve and explain better our results.
Thank you for your important observations, we tried to fit them in and improve the quality of our manuscript.

Reviewer 3 Report
I thank the authors for detailed answers. The authors have improved the quality of the paper. After clarification of the dimerization, I strongly suggest to add this "theory" in the paper, introduction section. Moreover since the first result is entitled "p53 forms dimers and binds to actin monomers and filaments".
Author Response
Summary: We thank the Editorial Board Members and Reviewers for their helpful and constructive comments. Additions or modifications made in direct responses are shown below. Changes made in response to those comments.
Reviewer Comments:
Answers to Reviewer 3.
After clarification of the dimerization, I strongly suggest to add this "theory" in the paper, introduction section. Moreover since the first result is entitled "p53 forms dimers and binds to actin monomers and filaments".
Thank you, we agree the importance of the Part of introduction to explain all required information what we want to interpret in our manuscript thus we tried to improve and explain better the p53 dimer formation in Part of Introduction.
Thank you for your important suggestion, we tried to fit it in and improve the quality of our manuscript.
